# Factors associated with pneumococcal carriage and density in infants and young children in Laos PDR

Eileen M. Dunne[1,2], Molina Choummanivong[3], Eleanor F. G. Neal[1,2], Kathryn Stanhope[1], Cattram D. Nguyen[1,2], Anonh Xeuatvongsa[4], Catherine Satzke[1,2,5], Vanphanom Sychareun[3], Fiona M. Russell[1,2]*

1 Infection and Immunity, Murdoch Children's Research Institute, Parkville, Victoria, Australia, 2 Department of Paediatrics, The University of Melbourne, Parkville, Victoria, Australia, 3 University of Health Sciences, Vientiane, Lao People's Democratic Republic, 4 Ministry of Health, Vientiane, Lao People's Democratic Republic, 5 Department of Microbiology and Immunology, Peter Doherty Institute for Infection and Immunity, The University of Melbourne, Parkville, Australia

* fmruss@unimelb.edu.au

**Data Availability Statement:** All relevant data are not publicly available because the ethics committee approved the study protocol, which specified the study objectives and how data would be used.

## Abstract

Nasopharyngeal carriage of *Streptococcus pneumoniae* (the pneumococcus) is a precursor to pneumococcal disease. Several host and environmental factors have been associated with pneumococcal carriage, however few studies have examined the relationship between host factors and pneumococcal carriage density. We sought to identify risk factors for pneumococcal carriage and density using data from cross-sectional pneumococcal carriage surveys conducted in the Lao People's Democratic Republic before and after the introduction of the 13-valent pneumococcal conjugate vaccine (PCV13). Nasopharyngeal swabs were collected infants from aged 5–8 weeks old (n = 999) and children aged 12–23 months (n = 1,010), pneumococci detected by quantitative PCR, and a risk factor questionnaire completed. Logistic and linear regression models were used to evaluate associations between participant characteristics and pneumococcal carriage and density. In infants aged 5–8 weeks, living in a household with two or more children under the age of five years (aOR 1.97; 95% CI 1.39–2.79) and low family income (aOR 1.64; 95% CI 0.99–2.72) were positively associated with pneumococcal carriage. For children aged 12–23 months, upper respiratory tract infection (URTI) symptoms (aOR 2.64; 95% CI 1.97–3.53), two or more children under five in the household (aOR 2.40; 95% CI 1.80–3.20), and rural residence (aOR 1.84, 95% CI 1.35–2.50) were positively associated with pneumococcal carriage. PCV13 vaccination was negatively associated with carriage of PCV13 serotypes (aOR 0.60; 95% CI 0.44–0.83). URTI symptoms (p < 0.001), current breastfeeding (p = 0.005), rural residence (p = 0.012), and delivery by Caesarean section (p = 0.035) were associated with higher mean pneumococcal density in pneumococcal carriers (both age groups combined). This study provides new data on pneumococcal carriage and density in a high disease burden setting in southeast Asia.

These restrictions apply to all data included in this manuscript. It is not consistent with our ethical permissions to share de-identified or aggregate versions of our data, as publicly available data could be used for purposes that were not specified in the protocol approved by the ethics committee, and therefore would be a breach of our ethics permissions. During the informed consent, the purpose of the study was explained to participants, and they were told how their data would be used. The use of these data for a new purpose that was not included in the approved study protocol would require additional ethical approval from the Lao PDR Ministry of Health National Ethics Committee for Health Research. Following approval, de-identified data would be made available. Additionally, this process is mindful of potential sensitivities regarding data from ethnic minorities. We have included contact information for the ethics committee via Dr Sengchanh Kounnavong, sengchanhkhounnavong@hotmail.com. We recommend that requests for data also be sent to the Principal Investigator, Prof. Fiona Russell (fmruss@unimelb.edu.au), so she can assist with the process.

**Funding:** The project was funded by Gavi, the Vaccine Alliance (https://www.gavi.org/) and the World Health Organization Western Pacific Regional Office (https://www.who.int/westernpacific), with support from the Victorian Government's Operational Infrastructure Support Program (https://www2.health.vic.gov.au/about/clinical-trials-and-research/operational-infrastructure-support). FMR was supported by a NHMRC Early Career and TRIP Fellowships (https://www.nhmrc.gov.au/). CS was supported by a NHMRC Career Development Fellowship (1087957) and a veski Inspiring Women Fellowship (https://www.veski.org.au/). EFGN holds an Australian Government Research Training Scholarship (https://www.education.gov.au/research-training-program). The funders had no role in study design, data collection and analysis, decision to publish, or preparation of the manuscript.

**Competing interests:** I have read the journal's policy and the authors of this manuscript have the following competing interests: EMD and CS have received research funding from Pfizer for an unrelated project. This does not alter our adherence to PLOS ONE policies on sharing data and materials.

## Introduction

The bacterium *Streptococcus pneumoniae* (the pneumococcus) is a significant global pathogen, and in 2015 was responsible for an estimated 317,300 deaths, most due to pneumonia, in children under five years old.[1] The majority of pneumococcal disease burden occurs in low- and middle- income countries (LMICs), believed to be due to poor access to health care and pneumococcal vaccines, and higher rates of risk factors such as HIV infection and malnutrition.[2] Other risk factors such as overcrowding contribute to the higher burden of pneumococcal disease in LMICs.[3]

Pneumococci are commonly carried in the nasopharynx of young children, with reported prevalence rates in LMICs ranging from 6 to 93%.[3] Although pneumococcal carriage is typically asymptomatic, it is a precursor for the development of pneumococcal disease and serves as the reservoir for this exclusively human pathogen.[4] High pneumococcal density (bacterial load) in the nasopharynx is associated with lower respiratory tract infections and pediatric pneumonia.[5]. In mouse models, high pneumococcal density in the nasopharynx increases transmission.[6]

Several host, socio-economic, and environmental factors are risk factors for pneumococcal carriage. These include day care attendance, living in a household with other young children, low socio-economic status, and symptoms of an upper respiratory tract infection (URTI).[7–11] In some studies, children in rural areas had higher odds of pneumococcal carriage compared to those living in urban settings.[12, 13] Co-infection with respiratory viruses, URTI symptoms, and low family income have been linked to higher pneumococcal density in children.[5, 9, 14, 15]

The Lao People's Democratic Republic (Lao PDR) is a lower-middle income country in East Asia. Lao PDR has a mortality rate of 67 per 1000 children under five, and a high burden of pneumonia in children. (https://www.gavi.org/country/lao-pdr/ [Accessed 13 June 2019]). The 13-valent pneumococcal conjugate vaccine (PCV13), which provides protection against 13 pneumococcal serotypes, was introduced into the national infant immunization schedule in Lao PDR in late 2013. Previously, we conducted nasopharyngeal carriage surveys before and two years after PCV13 introduction to evaluate the impact of the PCV program on pneumococcal carriage in two age groups: young children aged 12–23 months (the majority of whom were PCV-vaccinated in the post-PCV13 survey), and unvaccinated infants aged 5–8 weeks.[16] Following PCV13 introduction, the carriage prevalence of overall pneumococci did not change. Prevalence of PCV13-serotypes declined significantly in 12–23 month old children (from 33% to 20%; adjusted prevalence ratio 0.77 [95% CI 0.61–0.96], and there was some evidence of indirect effects in 5–8 week old infants (decline from 7% to 5%, adjusted prevalence ratio 0.74 [95% CI 0.43–1.27). Here, we describe a secondary analysis conducted using data from these cross-sectional surveys to identify demographic and household factors associated with carriage of pneumococci overall, PCV13 serotype carriage, and pneumococcal carriage density.

## Materials and methods

### Study design and participants

Cross-sectional nasopharyngeal carriage surveys were conducted from November 2013—February 2014 ("pre-PCV"), and November 2015—February 2016 ("post-PCV") as described previously.[16] Participants were enrolled in Vientiane, the capital of Lao PDR, and the rural Bolikhamxay Province. Urban participants were enrolled from maternal and child health centers during routine clinic visits. Rural participants were enrolled during maternal and child

health visits and visits to surrounding households. Inclusion criteria were age (5–8 week old infants and 12–23 month old children), temperature $\leq 37°C$ per axilla, and having lived in the area for at least three consecutive months. Previous receipt of PCV13 was an exclusion criteria for children and infants during the first survey, for infants aged 5–8 weeks for the second survey. Nasopharyngeal swabs were collected and stored according to WHO guidelines.[17] Study staff completed a questionnaire to collect data on demographics and potential risk factors (chosen based upon the literature) for each participant. These data included: sex, ethnicity (self-reported), urban or rural residence, the presence of URTI symptoms (including coryza, allergic rhinitis, or cough), antibiotic usage in the previous two weeks, exposure to household cigarette smoke, main source of cooking fuel (wood, charcoal, gas, electricity, kerosene), number and ages of other children in the household, family income (Kip/week), mode of delivery, and current breastfeeding status. A binary value for poverty was created using the $1.25 USD per day value international poverty line recommended by the World Bank from 2008–2015 (http://www.worldbank.org/). PCV13 immunization history was obtained from participant vaccination status cards or health center records.

The study was undertaken according to the protocol approved by the Lao PDR Ministry of Health National Ethics Committee for Health Research (061-NECHR), the Western Pacific Regional Office Ethics Review Committee, and The Royal Children's Hospital/Murdoch Children's Research Institute Human Research Ethics Committee (HREC 33177A/HREC 33177B). Written informed consent was obtained from the parent or guardian for all participants.

## Laboratory procedures

Samples were stored in ultra-low temperature freezers at the Lao-Oxford-Mahosot Hospital Wellcome Trust Research Unit (LOMWRU, Vientiane, Lao PDR) until shipment on dry ice to the Murdoch Children's Research Institute (Parkville, Australia) for microbiological analysis. Detection and quantification of pneumococci and pneumococcal serotyping were performed as previously described.[16] In brief, DNA extraction of 100 μl swab media was conducted using a MagNA Pure LC instrument (Roche) and pneumococci were detected and quantified by real-time quantitative PCR (qPCR) targeting the *lytA* gene.[18] A standard curve prepared using genomic DNA extracted from a reference isolate of *S. pneumoniae* (ATCC 6305) was used for quantification. The standard curve also served as a positive control, and extraction controls and no template controls were included as negative controls in each qPCR run.

Pneumococcal density data were reported in genome equivalents/ml (GE/ml). This unit approximates pneumococcal density with the assumption that each pneumococcal cell contains one genome (with a genome size of 2 Mb) and each genome contains one copy of the *lytA* gene, and an extraction volume of 0.1 ml swab media.

Pneumococcal-positive samples underwent molecular serotyping by microarray following a culture-amplification step on horse blood agar plates containing 5 μg/ml of gentamicin (Oxoid). DNA was extracted using a QIAcube HT instrument (Qiagen) and microarray performed using the Senti-SPv1.5 microarray (BUGS Bioscience).[19]

## Statistical analysis

Data entry and cleaning were conducted using EpiData version 3.1 and Stata version 15.1 as previously described.[16] Statistical analyses was performed using Stata 15.0. To evaluate the relationship between participant characteristics and pneumococcal carriage, unadjusted and adjusted odds ratios and 95% confidence intervals (CIs) were determined using logistic regression. Potential risk factors were examined separately for each age group for overall pneumococcal carriage. The following variables (shown in Table 1) were evaluated in the univariable

**Table 1. Characteristics of pneumococcal carriage survey participants in Lao PDR, conducted before and two years after PCV13 introduction.**

| Characteristics | | 5–8 week old infants (n = 999)[a] | 12–23 month old children (n = 1,010)[a] |
|---|---|---|---|
| **Survey, n (%)** | | | |
| | Pre-PCV13 | 498 (49.9) | 503 (49.8) |
| | Two years post-PCV13 | 501 (50.2) | 507 (50.2) |
| **Age. median (IQR)[b]** | | 6.7 weeks (6.5–7.0) | 16.6 months (15.6–20.0) |
| **Male sex, n (%)** | | n = 998 509 (51.0) | 484 (47.9) |
| **Ethnicity, n (%)** | | n = 998 | |
| | Lao Loum | 964 (96.6) | 963 (95.4) |
| | Lao Thung | 3 (0.3) | 13 (1.3) |
| | Hmong | 27 (2.7) | 30 (3.0) |
| | Other | 4 (0.4) | 4 (0.4) |
| **Rural residence, n (%)** | | 70 (7.0) | 470 (46.5) |
| **URTI[c] symptoms, n (%)** | | 162 (16.2) | 625 (61.9) |
| **Antibiotic use in the previous two weeks, n (%)** | | n = 997 29 (2.9) | n = 1,007 405 (40.2) |
| **Exposure to household cigarette smoke, n (%)** | | n = 996 366 (36.8) | n = 1,007 440 (43.7) |
| **Primary household fuel source, n (%)** | | | n = 1,007 |
| | wood | 164 (16.4) | 227 (27.5) |
| | charcoal | 510 (51.1) | 482 (47.8) |
| | kerosene | 1 (0.1) | 1 (0.1) |
| | gas | 179 (19.7) | 153 (15.2) |
| | electricity | 145 (15.4) | 95 (9.4) |
| **Two or more children < 5 years in the household, n (%)** | | n = 998 409 (41.0) | 349 (34.6) |
| **Below poverty line[d], n (%)** | | 109 (10.6) | 70 (6.9) |
| **Delivered by Caesarean section, n (%)** | | 238 (23.8) | 179 (17.7) |
| **Current breastfeeding, n (%)** | | n = 998 844 (84.6) | n = 1,009 201 (19.9) |
| **PCV13 vaccinated[e], n (%)** | | 0 (0.0) | n = 1,000 448 (44.8) |

[a]If participant data were missing for an individual variable, results for that variable were excluded and the denominator used was the number of participants with data available.

[b]IQR = interquartile range

[c]URTI = upper respiratory tract infection

[d]Defined as $1.25 USD/day

[e]Received 2 or 3 doses of PCV13

analysis: sex, location (urban or rural), ethnicity, the presence of upper URTI symptoms, antibiotic use in the previous two weeks, exposure to cigarette smoke, primary source of fuel, two or more children under the age of five years in the household, family income below the poverty line, mode of delivery, breastfeeding status, and survey year. Primary source of fuel was categorized into biofuels (wood and charcoal) or non-biofuels (electricity, gas, and kerosene) and ethnicity data were reclassified as Lao Loum or other prior to analysis. For 12–23 month old children, PCV13 vaccination status was also assessed, however survey year was omitted due to co-linearity with PCV13 status. Adjusted odds ratios and 95% CI were calculated using

multivariable logistic regression models included variables selected *a priori* based on the literature (URTI symptoms, two or more children under the age of five years in the household, poverty) and any variables with p < 0.2 by univariable analysis. Similar analyses were also undertaken for PCV13 serotype carriage for the 12–23 month old age group. When serotyping results were not available from a sample due to technical issues, that sample was excluded from serotype-specific analyses.

Pneumococcal density data were $log_{10}$ transformed prior to analysis. Potential relationships between potential risk factors described above and pneumococcal density we evaluated using linear regression. This analysis combined both age groups and was restricted to pneumococcal carriers. Multivariable linear regression models included URTI symptoms (selected *a priori*) plus variables with p < 0.2 by univariable analysis. Results were reported as the unadjusted and adjusted coefficients (the difference in mean density) with 95% CIs and p values.

## Results

Table 1 summarizes characteristics of the 2,009 study participants. Generally, participant characteristics were similar between age groups with a few exceptions: very few of the 5–8 week old infants lived in rural areas, URTI symptoms and recent antibiotic use was much more common in 12–23 month old children, and breastfeeding was more common in the young infants. In the post-PCV survey, 448/497 (90.1%) of participants aged 12–23 months had received two or more doses of PCV13. Information on PCV13 immunization history was not available for ten participants.

One swab was excluded due to technical issues, and 9 swabs that were positive for pneumococcus were not able to be serotyped. For 5–8 week old infants, 157/999 (15.7%; 95% CI 13.5–18.1) carried pneumococcus, and 58/995 (5.8%; 95% CI 4.5–7.5) carried a PCV13 serotype. For the 12–23 month old children, 511/1009 (50.6%; 95% CI 47.5–53.8) carried pneumococcus, and 265/1005 (26.4%; 95% CI 23.7–29.2) carried a PCV13 serotype. Pneumococcal density was variable among carriers, ranging from 2.9 to 8.2 $log_{10}$ GE/ml. The median carriage density was 5.8 $log_{10}$ GE/ml (range 3.5–8.2; IQR 5.1–6.6) in 5–8 week old infants and 5.8 $log_{10}$ GE/ml (range 2.9–8.1; IQR 5.0–6.5) in 12–23 month old children.

Associations between participant characteristics and overall pneumococcal carriage for each age group are shown in Tables 2 and 3. For 5–8 week old infants, living in a household with two or more children under the age of five years (aOR 1.97; 95% CI 1.39–2.79) was positively associated with pneumococcal carriage, and there was some evidence for a positive association with having a family income below the poverty line (aOR 1.64; 95% CI 0.99–2.71). Living in a household with two or more children under five was positively associated with pneumococcal carriage in children aged 12–23 months (aOR 2.40; 95% CI 1.80–3.20), along with rural residence (aOR 1.84; 95% CI 1.35–2.50) and having URTI symptoms (aOR 2.64; 95% CI 1.97–3.53). URTI symptoms were significantly more common in rural children (336/ 470, 71.5%) compared to children from urban areas (289/540, 53.7%, p < 0.001, chi-squared test).

Carriage of PCV13 serotypes was also examined, as these serotypes are more likely to cause invasive pneumococcal disease, and specifically targeted by the PCV immunization program. Due to the relatively small number of 5–8 week old infants who carried a PCV13 serotype, this analysis was only conducted for 12–23 month old children (Table 4). Factors identified to be positively associated with PCV13 serotype carriage were consistent with those associated with overall pneumococcal carriage (rural residence, presence of URTI symptoms, and living in a household with two or more children under five). Additionally, previous PCV13 vaccination was associated with reduced odds of PCV13 serotype carriage (aOR 0.60; 95% CI 0.44–0.83).

**Table 2. Univariable and multivariable analysis of participant characteristics associated with pneumococcal carriage in infants aged 5–8 weeks (n = 999).**

| Variable | Pneumococcal carriage n/N (%) | Unadjusted odds ratio (95% CI) | P value | Adjusted odds ratio (95% CI)[a] | P value |
|---|---|---|---|---|---|
| **Sex** | | | | | |
| female | 75/489 (15.3) | ref | | | |
| male | 82/509 (16.1) | 1.06 (0.75–1.49) | 0.738 | | |
| **Ethnicity** | | | | | |
| Lao Loum | 148/964 (15.4) | ref | | | |
| other[b] | 9/34 (27) | 1.98 (0.91–4.34) | 0.086 | 1.65 (0.72–3.76) | 0.233 |
| **Residence type** | | | | | |
| urban | 139/929 (15.0) | ref | | | |
| rural | 18/70 (26) | 1.97 (1.12–3.46) | 0.019 | 1.56 (0.85–2.84) | 0.152 |
| **URTI[c] symptoms** | | | | | |
| no | 127/837 (15.2) | ref | | | |
| yes | 30/162 (18.5) | 1.27 (0.82–1.97) | 0.285 | 1.17 (0.74–1.85) | 0.493 |
| **Antibiotic use in the previous 2 weeks** | | | | | |
| no | 154/814 (15.9) | ref | | | |
| yes | 3/29 (10) | 0.61 (0.18–2.04) | 0.422 | | |
| **Exposure to household cigarette smoke** | | | | | |
| no | 95/630 (15.1) | ref | | | |
| yes | 60/366 (16.4) | 1.10 (0.78–1.57) | 0.581 | | |
| **Primary household fuel source[d]** | | | | | |
| non-biofuel | 49/325 (15.1) | ref | | | |
| biofuel | 108/674 (16.0) | 1.08 (0.74–1.55) | 0.700 | | |
| **2 or more children <5y in the household** | | | | | |
| no | 70/589 (11.9) | ref | | | |
| yes | 86/409 (21.0) | 1.97 (1.17–2.79) | <0.001 | 1.97 (1.39–2.79) | <0.001 |
| **Family income** | | | | | |
| above poverty line | 131/893 (14.7) | ref | | | |
| below poverty line | 26/106 (24.5) | 1.89 (1.17–3.05) | 0.009 | 1.64 (0.99–2.72) | 0.055 |
| **Mode of delivery** | | | | | |
| vaginal | 129/761 (17.0) | ref | | | |
| Caesarean | 28/238 (11.8) | 0.65 (0.42–1.01) | 0.056 | 0.69 (0.44–1.09) | 0.108 |
| **Currently breastfeeding** | | | | | |
| no | 24/130 (15.6) | ref | | | |
| yes | 133/844 (15.8) | 1.01 (0.63–1.62) | 0.957 | | |
| **Survey** | | | | | |
| pre-PCV13 | 71/498 (14.3) | | | | |
| 2 years post-PCV13 | 86/501 (17.2) | 1.25 (0.89–1.76) | 0.207 | | |

[a]Pseudo R$^2$ = 0.084

[b]other includes Lao Thung, Hmong, and other

[c]URTI = upper respiratory tract infection

[d]biofuel = wood or charcoal; non-biofuel = gas, kerosene, or electricity

To examine factors associated with PCV13 serotype carriage prior to PCV13 introduction, we conducted an analysis only including children from the pre-PCV survey. Results were consistent with those from the analysis of both surveys presented in Table 4: following multivariable analysis, rural living (aOR 1.70; 95% CI 1.08–2.68; p = 0.021), the presence of URTI symptoms (aOR 2.44; 95%CI 1.46–4.07, p = 0.001), and living in a household with two or more children

**Table 3. Univariable and multivariable analysis of participant characteristics associated with pneumococcal carriage in children aged 12–23 months (n = 1,009).**

| Variable | Pneumococcal carriage n/N (%) | Unadjusted odds ratio (95% CI) | P value | Adjusted odds ratio (95% CI)[a] | P value |
|---|---|---|---|---|---|
| **Sex** | | | | | |
| female | 278/526 (52.9) | ref | | | |
| male | 233/483 (48.2) | 0.83 (0.65–1.06) | 0.143 | 0.79 (0.60–1.03) | 0.080 |
| **Ethnicity** | | | | | |
| Lao Loum | 480/962 (49.9) | ref | | | |
| other[b] | 31/47 (66) | 1.94 (1.05–3.60) | 0.034 | 1.07 (0.54–2.11) | 0.852 |
| **Residence type** | | | | | |
| urban | 223/540 (41.3) | ref | | | |
| rural | 288/469 (61.4) | 2.26 (1.76–2.91) | <0.001 | 1.84 (1.35–2.50) | <0.001 |
| **URTI[c] symptoms** | | | | | |
| no | 135/385 (35.1) | ref | | | |
| yes | 376/624 (60.3) | 2.81 (2.16–3.66) | <0.001 | 2.64 (1.97–3.53) | <0.001 |
| **Antibiotic use in the previous 2 weeks** | | | | | |
| no | 293/602 (48.7) | ref | | | |
| yes | 218/404 (54.0) | 1.24 (0.96–1.59) | 0.100 | 0.91 (0.69–1.21) | 0.523 |
| **Exposure to household cigarette smoke** | | | | | |
| no | 276/567 (48.7) | | | | |
| yes | 234/439 (53.0) | 1.20 (0.94–1.54) | 0.146 | 0.90 (0.68–1.20) | 0.474 |
| **Primary household fuel source[d]** | | | | | |
| non-biofuel | 100/249 (40.2) | ref | | | |
| biofuel | 410/758 (54.1) | 1.76 (1.31–2.35) | <0.001 | 1.24 (0.88–1.73) | 0.214 |
| **children <5y in the household** | | | | | |
| 1 | 291/660 (44.1) | ref | | | |
| 2 or more | 220/349 (63.0) | 2.16 (1.66–2.82) | <0.001 | 2.40 (1.80–3.20) | <0.001 |
| **Family income** | | | | | |
| above poverty line | 471/939 (50.2) | ref | | | |
| below poverty line | 40/70 (57) | 1.32 (0.81–2.16) | 0.261 | 0.96 (0.56–1.64) | 0.887 |
| **Mode of delivery** | | | | | |
| vaginal | 437/830 (52.7) | ref | | | |
| Caesarean | 74/179 (41.3) | 0.63 (0.46–0.88) | 0.006 | 0.80 (0.56–1.15) | 0.227 |
| **Currently breastfeeding** | | | | | |
| no | 403/807 (49.9) | ref | | | |
| yes | 108/201 (53.7) | 1.16 (0.85–1.59) | 0.336 | | |
| **PCV13 vaccination history** | | | | | |
| 0 or 1 dose | 304/551 (55.2) | ref | | | |
| 2 or 3 doses | 202/448 (45.1) | 0.67 (0.52–0.86) | 0.002 | 0.82 (0.62–1.09) | 0.168 |

[a]Pseudo $R^2$ = 0.097

[b]other includes Lao Thung, Hmong, and other

[c]URTI = upper respiratory tract infection

[d]biofuel = wood or charcoal; non-biofuel = gas, kerosene, or electricity

under five (aOR 2.59; 95%CI 1.72–3.89; p < 0.001) were associated with PCV13 serotype carriage in the pre-PCV13 survey.

Next, we examined factors associated with pneumococcal carriage density in children and infants positive for pneumococcal carriage. As no differences in pneumococcal density were observed between the two age groups surveyed (p = 0.574, t-test), data were pooled for analysis

**Table 4. Univariable and multivariable analysis of participant characteristics associated with carriage of PCV13 serotypes in children aged 12–23 months (n = 1,005).**

| Variable | Pneumococcal carriage n/N (%) | Unadjusted odds ratio (95% CI) | P value | Adjusted odds ratio (95% CI)[a] | P value |
|---|---|---|---|---|---|
| **Sex** | | | | | |
| female | 144/524 (27.5) | ref | | | |
| male | 121/482 (25.1) | 0.88 (0.67–1.17) | 0.383 | | |
| **Ethnicity** | | | | | |
| Lao Loum | 246/958 (25.7) | ref | | | |
| Other[b] | 19/47 (40) | 1.96 (1.08–3.58) | 0.028 | 1.22 (0.64–2.31) | 0.552 |
| **Residence type** | | | | | |
| urban | 102/539 (18.9) | ref | | | |
| rural | 163/466 (35.0) | 2.30 (1.73–3.01) | <0.001 | 1.84 (1.32–2.56) | <0.001 |
| **URTI[c] symptoms** | | | | | |
| no | 61/384 (15.9) | ref | | | |
| yes | 204/621 (32.9) | 2.59 (1.88–3.57) | <0.001 | 2.15 (1.52–3.04) | <0.001 |
| **Antibiotic use in the previous 2 weeks** | | | | | |
| no | 148/601 (24.6) | ref | | | |
| yes | 117/401 (29.2) | 1.26 (0.95–1.68) | 0.110 | 1.00 (0.74–1.37) | 0.974 |
| **Exposure to household cigarette smoke** | | | | | |
| no | 143/565 (25.3) | | | | |
| yes | 121/437 (27.7) | 1.13 (0.85–1.50) | 0.397 | | |
| **Primary household fuel source[d]** | | | | | |
| non-biofuel | 48/247 (19.4) | ref | | | |
| biofuel | 217/756 (28.7) | 1.67 (1.18–2.37) | 0.004 | 1.10 (0.73–1.64) | 0.655 |
| **children <5y in the household** | | | | | |
| 1 | 140/657 (21.3) | ref | | | |
| 2 or more | 125/348 (35.9) | 2.07 (1.55–2.76) | <0.001 | 2.27 (1.67–3.08) | <0.001 |
| **Family income** | | | | | |
| above poverty line | 241/935 (25.8) | ref | | | |
| below poverty line | 24/70 (34) | 1.50 (0.90–2.51) | 0.121 | 1.07 (0.62–1.86) | 0.807 |
| **Mode of delivery** | | | | | |
| vaginal | 221/826 (26.7) | ref | | | |
| Caesarean | 44/179 (24.6) | 0.89 (0.61–1.30) | 0.550 | | |
| **Currently breastfeeding** | | | | | |
| no | 206/804 (25.6) | ref | | | |
| yes | 59/200 (29.5) | 1.21 (0.86–1.71) | 0.266 | | |
| **PCV13 vaccination history** | | | | | |
| 0 or 1 dose | 176/550 (32.0) | ref | | | |
| 2 or 3 doses | 87/445 (19.6) | 0.52 (0.38–0.69) | <0.001 | 0.60 (0.44–0.83) | 0.002 |

[a]Pseudo $R^2$ = 0.084

[b]other includes Lao Thung, Hmong, and other

[c]URTI = upper respiratory tract infection

[d]biofuel = wood or charcoal; non-biofuel = gas, kerosene, or electricity

by linear regression, with results shown in Table 5. Having URTI symptoms was associated with increased pneumococcal density, with an adjusted coefficient (difference between means) of 0.34 $\log_{10}$ GE/ml (95% CI 0.17–0.50; p < 0.001). Rural residence (0.20 $\log_{10}$ GE/ml; 95% CI 0.05–0.36; p = 0.012), delivery via Caesarean section (0.22 $\log_{10}$ GE/ml; 95% CI 0.02–0.43; p = 0.035) and current breastfeeding (0.24 $\log_{10}$ GE/ml; 95% CI 0.07–0.41; p = 0.005) were also

**Table 5. Univariable and multivariable analysis of factors associated with pneumococcal density in infants and young children positive for pneumococcal carriage (n = 668).**

| Variable (n) | Mean (range) pneumococcal density, $\log_{10}$ GE/ml | Unadjusted coefficient[a] (95% CI) | P value | Adjusted coefficient[b] (95% CI) | P value |
|---|---|---|---|---|---|
| **Sex** | | | | | |
| female (353) | 5.77 (3.19–8.17) | | | | |
| male (315) | 5.69 (2.85–7.73) | -0.08 (-0.23–0.07) | 0.308 | | |
| **Ethnicity** | | | | | |
| Lao Loum (628) | 5.73 (2.85–8.17) | | | | |
| other (40) | 5.84 (4.10–7.73) | 0.11 (-0.21–0.44) | 0.491 | | |
| **Residence type** | | | | | |
| urban (362) | 5.67 (3.31–8.17) | | | | |
| rural (306) | 5.81 (2.85–8.12) | 0.14 (-0.01–0.29) | 0.071 | 0.20 (0.05–0.36) | 0.012 |
| **URTI[c] symptoms** | | | | | |
| no (262) | 5.62 (3.31–8.17) | | | | |
| yes (406) | 5.81 (2.85–8.07) | 0.18 (0.03–0.34) | 0.023 | 0.34 (0.17–0.50) | <0.001 |
| **Antibiotic use in the previous 2 weeks** | | | | | |
| no (447) | 5.74 (2.85–8.17) | | | | |
| yes (221) | 5.72 (3.19–8.07) | -0.02 (-0.19–0.14) | 0.752 | | |
| **Exposure to household cigarette smoke** | | | | | |
| no (371) | 5.74 (2.85–8.12) | | | | |
| yes (294) | 5.73 (3.19–8.17) | 0.02 (-0.17–0.14) | 0.879 | | |
| **Primary household fuel source[d]** | | | | | |
| non-biofuel (149) | 5.66 (3.19–8.12) | | | | |
| biofuel (518) | 5.76 (2.85–8.17) | 0.10 (-0.08–0.29) | 0.274 | | |
| **Children <5y in the household** | | | | | |
| 1 (361) | 5.74 (3.24–8.12) | | | | |
| 2 or more (306) | 5.72 (2.85–8.17) | -0.03 (-0.18–0.13) | 0.735 | | |
| **Family income** | | | | | |
| above poverty line (602) | 5.73 (2.85–8.17) | | | | |
| below poverty line (66) | 5.74 (3.31–7.72) | 0.01 (-0.25–0.26) | 0.950 | | |
| **Mode of delivery** | | | | | |
| vaginal (566) | 5.70 (2.85–8.17) | | | | |
| Caesarean (102) | 5.90 (3.90–8.12) | 0.20 (-0.02–0.41) | 0.069 | 0.22 (0.02–0.43) | 0.035 |
| **Currently breastfeeding** | | | | | |
| no (427) | 5.67 (2.85–8.07) | | | | |
| yes (241) | 5.85 (3.50–8.17) | 0.17 (0.02–0.33) | 0.030 | 0.24 (0.07–0.41) | 0.005 |
| **PCV13 vaccine history** | | | | | |
| 0 or 1 dose (460) | 5.62 (2.85–8.17) | | | | |
| 2 or 3 doses (205) | 5.97 (3.31–8.07) | 0.34 (0.18–0.50) | <0.001 | -0.10 (-0.34–0.14) | 0.435 |
| **Survey year** | | | | | |
| pre-PCV13 (351) | 5.49 (2.85–7.62) | | | | |
| post-PCV13 (317) | 6.00 (3.31–8.17) | 0.51 (0.36–0.66) | <0.001 | 0.64 (0.42–0.87) | <0.001 |

[a]coefficient is the difference between means determined by linear regression

[b]adjusted $R^2$ = 0.103

[c]URTI = upper respiratory tract infection

[d]biofuel = wood or charcoal; non-biofuel = gas, kerosene, or electricity

associated with higher pneumococcal density. As reported previously, pneumococcal densities were higher in the post-PCV13 survey compared with the pre-PCV13 survey (0.64 $\log_{10}$ GE/ml; 95% CI 0.42–0.87; p < 0.001).[16]

## Discussion

Southeast Asia has one of the highest estimated incidence of pneumococcal disease in children in the world.[1] However, relatively few studies have examined risk factors for pneumococcal carriage, a precursor to pneumococcal disease, in this setting. We examined carriage in infants aged 5–8 weeks old prior to their first dose of PCV13, as unvaccinated neonates and young infants are particularly vulnerable to invasive bacterial infections, including those caused by pneumococcus.[20] In this age group, having and two or more children under five in the household and low family income were risk factors for pneumococcal carriage. In Lao PDR children aged 12–23 months, URTI symptoms, two or more children under five years old in the household, and rural residence were associated with pneumococcal carriage. Results from our study were similar with other pneumococcal carriage studies conducted in the region. In Vietnam, age and day care attendance were associated with pneumococcal carriage in children under five.[21] In a longitudinal study of infants in Thailand followed from birth, maternal smoking and presence of other young children in the household were associated with earlier pneumococcal colonization.[22] Living in a household containing two or more children under five years old and URTI symptoms were identified as risk factors for pneumococcal carriage in Indonesian children aged 12–24 months.[9] Results from our study advance upon previous work by assessing risk factors for pneumococcal carriage in a unique population within which PCV has been recently introduced.

Lao PDR is a diverse nation, with 49 distinct ethnic groups, and historically, ethnic minority groups have had poorer health status, including increased childhood mortality and lower vaccination rates.[23] As our study was conducted in the capital city and a nearby province, relatively few participants belonging to ethnic minorities were enrolled, and therefore results are not representative of the overall population of Lao PDR. Nearly half of 12–23 month old participants, compared with 7% of 5–8 week old infants, lived in rural areas; this may explain why rural residence was identified as a risk factor only in the older age group. Consistent with a previous study on respiratory infection burden in Lao PDR, children in rural areas were more likely to have UTRI symptoms.[24]

Risk factors for PCV13 carriage were similar to those associated with overall pneumococcal carriage, and as expected, children who received two or three doses of PCV13 had reduced odds of carrying a PCV13 serotype. In later years following PCV13 introduction, carriage of PCV13 serotypes may become associated with contact with under-vaccinated communities and/or interactions with older, unvaccinated children and adolescents.[25]

In our study, pneumococcal carriage was not associated with recent antibiotic use; this may be due to the unreliability of parent-reported data on antibiotic use, which has been previously documented in Lao PDR, and high rates of antimicrobial resistance in pneumococci.[26] We previously reported that over 70% of pneumococcal-positive carriage samples harbored at least one antimicrobial resistance gene.[16] Results from studies examining cigarette exposure as a risk factor for pneumococcal carriage have reported varying results, and we did not find an association in our study population.[9, 27–29] However, exposure to cigarette smoke can be difficult to determine without monitoring, and levels of smoke exposure and maternal smoking were not assessed in our study. Indoor air pollution caused by the use of solid fuels such as wood and charcoal as cooking fuel is a risk factor for pneumonia in children.[30] In our study,

biofuels were associated with increased pneumococcal carriage in 12–23 month old children by univariable analysis but not following multivariable analysis.

High pneumococcal density in the nasopharynx is associated with respiratory infections including pneumonia in children.[5, 31, 32] Additionally, pneumococcal density in the nasopharynx has been investigated as a potential diagnostic tool for pediatric pneumonia.[31] As such, it is important to understand what underlying host factors may influence pneumococcal carriage density, and our study provides new data on this topic. Consistent with other studies, individual pneumococcal carriage density varied widely.[33,34] We identified several factors associated with increased pneumococcal density. The presence of URTI symptoms was positively associated with pneumococcal density in our study, consistent with results from community carriage surveys in young children in Indonesia and Belgium.[9, 35] Living in a rural area was associated with increased pneumococcal density; this may be related to socio-economic and environmental differences between urban and rural households. Pneumococcal density was higher in the post-PCV carriage survey, but as this was the case for both PCV13 serotypes and non-PCV13 serotypes, this was not attributed to PCV.[16] We have performed pneumococcal density studies before and after PCV in Fiji, Mongolia, and Lao PDR. Similar to our findings in Lao PDR, density of both PCV13 and non-PCV13 serotypes was higher in children following PCV13 introduction in Mongolia.[36] In contrast, density of both vaccine serotypes and non-vaccine serotypes declined in children in Fiji following the introduction of PCV10.[34] In an experimental human pneumococcal challenge model, adults who received PCV13 had lower pneumococcal density compared with those who received a control vaccine.[37] To date, there is not consistent evidence of an effect of PCV on pneumococcal density in children. Variations in density observed in cross-sectional studies with carriage assessed during different years may be temporal and/or related to as number of unmeasured factors.

Two previously unreported factors were linked with higher pneumococcal density: delivery by Caesarean section and current breastfeeding. Unlike URTI symptoms and rural living, both of these factors were not associated with the likelihood of pneumococcal carriage, but rather with increased pneumococcal loads in pneumococcal carriers. Mode of delivery has been found to significantly affect the development of the nasopharyngeal microbiome through the first six months of life.[38] Caesarean birth was associated with a high abundance of Streptococcus from early life in a nasopharyngeal microbiome study of Dutch infants.[39] It is possible that Caesarean delivery may influence microbiome development in a manner that supports pneumococcal growth, however this finding requires further investigation. We did not collect detailed breastfeeding data on exclusive vs. non-exclusive breastfeeding, formula use, and duration of exclusive breastfeeding, so the relationship between breastfeeding and increased pneumococcal density should be interpreted with caution. A study comparing the nasopharyngeal microbiome of exclusively breast-fed infants with formula-fed infants found differences in bacterial community composition at six weeks of age that resolved by six months.[40] The generation R study examining risk factors for pneumococcal carriage in Dutch infants found no association between pneumococcal carriage and duration of breast-feeding or exclusive breast-feeding at 1.5, 6 or 14 months of age.[11] Only 20% of participants in the older age group reported current breastfeeding, and it is possible that there may be underlying socioeconomic and other differences between the breastfeeding group and those in the same age group who were no longer breastfed that were not captured therefore not included in the adjusted analysis. Alternatively, it is possible that breastfeeding may negatively affect bacterial species such as *Staphylococcus aureus* that are competitors of pneumococcus, therefore facilitating pneumococcal growth, although this is speculative.[41]

Viral testing was not conducted in our study as we recruited generally healthy, afebrile children from the community. However, > 60% of participants aged 12–23 months had URTI

symptoms. Several studies have identified positive associations between respiratory viruses and pneumococcal carriage and density in the nasopharynx. In children attending day-care in Portugal, pneumococcal density was significantly higher in children who tested positive for a respiratory virus.[14] In a UK clinical trial, children who received the live attenuated influenza vaccine had significantly higher pneumococcal density compared with controls when assessed 28 days after vaccination.[42] In rural children in Peru, respiratory virus detection was positively associated with pneumococcal density.[43] A surveillance study on influenza-like illness in households in Vientiane, Lao PDR found that pneumococcus, influenza virus, and parainfluenza virus were the most common pathogens detected in children aged 0–4 years, and that co-detection of pneumococcus and respiratory viruses was common.[44] Combined, these findings suggest that children in our study with high pneumococcal density and URTI symptoms might have been co-infected with a respiratory virus. However, asymptomatic viral co-infection may also influence pneumococcal density: a study in American children found that pneumococcal densities were higher when a respiratory virus was detected, even in the absence of URTI symptoms.[15]

We did not evaluate HIV infection status or malnutrition, factors that may influence pneumococcal carriage, in our study. Lao PDR has a low prevalence of HIV (estimated at 0.2% of 18 to 49 year olds in 2013) and therefore we assume similar low prevalence in our participants.[45] Childhood malnutrition is a major public health issue in Lao PDR. In 2015, the prevalence of stunting, which is an indicator of chronic malnutrition, in children under five was 36%.[23] Malnutrition was likely common among the study population, and has been identified as a risk factor for pneumococcal carriage in other studies.[46, 47]

In summary, we have identified host and socio-economic risk factors for pneumococcal carriage and density in infants and young children in Lao PDR. Our findings highlight links between URTI infections, pneumococcal carriage, and pneumococcal density, and also identify children living in rural areas as having increased risk of pneumococcal carriage.

## Acknowledgments

We thank study participants and their families, and the University of Health Sciences project staff for enrolment and sample collection. We thank the Lao-Oxford-Mahosot Hospital Wellcome Trust Research Unit for sample processing, and the pneumococcal microbiology lab at the Murdoch Children's Research Institute and the Bacterial Microarray Group at St George's, University of London for microbiological analysis.

## Author Contributions

**Conceptualization:** Anonh Xeuatvongsa, Vanphanom Sychareun, Fiona M. Russell.

**Data curation:** Molina Choummanivong, Eleanor F. G. Neal, Kathryn Stanhope.

**Formal analysis:** Eileen M. Dunne.

**Funding acquisition:** Fiona M. Russell.

**Investigation:** Molina Choummanivong, Kathryn Stanhope.

**Methodology:** Eileen M. Dunne, Eleanor F. G. Neal, Cattram D. Nguyen, Catherine Satzke, Vanphanom Sychareun.

**Resources:** Anonh Xeuatvongsa.

**Supervision:** Catherine Satzke, Fiona M. Russell.

**Writing – original draft:** Eileen M. Dunne.

**Writing – review & editing:** Eleanor F. G. Neal, Catherine Satzke, Fiona M. Russell.

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
