## [Decision Letter · Decision Letter 0]

12 Aug 2019

PONE-D-19-19269

Factors associated with pneumococcal carriage and density in infants and young children in Laos PDR

PLOS ONE

Dear Dr Dunne,

Thank you for submitting your manuscript to PLOS ONE. After careful consideration, we feel that it has merit but does not fully meet PLOS ONE’s publication criteria as it currently stands. Therefore, we invite you to submit a revised version of the manuscript that addresses the points raised during the review process.

We would appreciate receiving your revised manuscript by Sep 26 2019 11:59PM. To enhance the reproducibility of your results, we recommend that if applicable you deposit your laboratory protocols in protocols.io, where a protocol can be assigned its own identifier (DOI) such that it can be cited independently in the future. For instructions see: http://journals.plos.org/plosone/s/submission-guidelines#loc-laboratory-protocols

We look forward to receiving your revised manuscript.

Kind regards,

Jean-San Chia

Academic Editor

PLOS ONE

Journal Requirements:

I have read the journal's policy and the authors of this manuscript have the following competing interests: EMD and CS have received research funding from Pfizer for an unrelated project.

Additional Editor Comments:

Your article has been carefully reviewed by three experts including two physician scientists, one in Pediatrician and one in infectious diseases. They have provided both instructive suggestions and constructive opinions for you to revise this manuscript in a timely manner

Reviewers' comments:

Reviewer's Responses to Questions

**Comments to the Author**

1. Is the manuscript technically sound, and do the data support the conclusions?

Reviewer #1: Partly

Reviewer #2: Partly

Reviewer #3: Yes

2. Has the statistical analysis been performed appropriately and rigorously? 

Reviewer #1: Yes

Reviewer #2: No

Reviewer #3: Yes

3. Have the authors made all data underlying the findings in their manuscript fully available?

Reviewer #1: Yes

Reviewer #2: No

Reviewer #3: Yes

4. Is the manuscript presented in an intelligible fashion and written in standard English?

Reviewer #1: Yes

Reviewer #2: No

Reviewer #3: Yes

5. Review Comments to the Author

Reviewer #1: In this manuscript, the authors analyzed the relationship between host factors and pneumococcal carriage density in the young children in the Lao People's Democratic Republic. They provide evidence that some host factors, including living environments, areas, socio-economic factors and URTI infections, would influence the pneumococcal carriage and density in young children. These data provide some information on pneumococcal carriage and density in a high pneumococcal disease burden in Lao, but some important information should be provided more clearly. The details of my concerns are listed as follows.

1. To quantify the bacterial density, the authors use the unit of “genome equivalents per ml (GE/ml)” to represent the bacterial density in this study, but little information about the unit of GE/ml was obtained from this manuscript. The authors should describe how to detect the “genome equivalents per ml” in more detail and tell us what the volume unit represent. They should also provide evidence or reference that the data of “genome equivalents per ml” can represent the absolute number of pneumococci isolated from the nasopharyngeal swab samples. The distribution of bacteria number isolated form the clinical samples are always wide. The result of the pneumococcal density showed in this study is only arranged from 5.49 to 6.00 (log10 GE/ml), which implied the limitation of the quantification assay. Therefore, more detailed description and evidence should be provided to support the presenting data.

2. PVC vaccination will significantly reduce the carriage of PCV13 serotypes (Table 4). Therefore, the carriage of pneumococci or PCV13 serotypes in the group of pre-vaccination should be specifically analyzed.

3. Some results described in the abstract, in the text and showed in the able are not consistent, such as the association of living in a household with two or more children under the age of five years (infants aged 5 - 8 weeks) and pneumococcal carriage. Please check that carefully.

Reviewer #2: The authors identified risk factors for pneumococcal carriage and density using data from cross-sectional pneumococcal carriage surveys conducted in the Lao People's Democratic Republic before and after the introduction of the 13-valent pneumococcal conjugate vaccine (PCV13). The authors found that in the age group of 5-8 weeks, having and two or more children under five in the 233 household and low family income were risk factors for pneumococcal carriage. Children aged 12 - 23 months, URTI symptoms, two or more children under five years old in the household, and rural residence were associated with pneumococcal carriage. URTI symptoms, current breastfeeding, rural residence, and delivery by Caesarean section were associated with higher mean pneumococcal density in pneumococcal carriers.

I have some major concerns.

1. The authors have published the other article using the same cohort. (Vaccine. 404 2019;37(2):296-305.) The author should clearly state the difference between the two studies.

2. The pneumococcal density was measured using real-time PCR according to the reference (J Clin Microbiol. 2007;45(8):2460-6.). However, in the initial study by Carvalho, the authors used the real-time PCR to detect rather than quantify the pneumococcus. Several studies are using the same method to quantify the pneumococcus. However, there are augment that measuring the pneumococcus DNA alone is likely to be confounded by variable production and sampling of nasopharyngeal mucous. (Sci Rep. 2018 Jul 23;8(1):11030.). I am wondering in the present study whether the pneumococcal density adjusted for the concentration of human DNA present? If "no", the authors might consider to debate why the adjustment is not necessary.

There also some additional comments.

1. If the data can be delinked, what is the ethical concerns that the data can't be publicly available?

2. Page 2 Line 30. The variable low family income seems statistically not significant.

3. Page 4 Line 73-75. The statement is statistically not significant.

4. Study design and participant section. In what healthcare setting were those participant enrolled? Were the participant of both age group enrolled in the same healthcare setting?

5. page 6, Line 107-110. The author might describe more detail about the method of the realtime PCR. What is used as a standard curve? How the density calculate? What is the detection limits? Did the experiment include positive control or negative control of the swab? Or did the Real-time PCR included positive or negative control?

6. Page 7, Line 132. Is there a preset cutoff for the statistical significance?

7. Page 7, Line 144. According to table 1, there are 507 participants of the age group 12-23 months in the two years post-PCV 13 . Why the denominator is 497 in Line 144 ?

8. Table 1. why the denominator varies in different rows?

9. if the participant of both age group enrolled in the same healthcare setting, it is weird that there are large differences of the distribution of the rural residence and the URTI symptoms.

10. Page 8, Line 155-158. Why are the denominators different? 999 or 995, and 1009 or 1005?

11. Page 9, Line 164-165. The statement is statistically not significant.

12. Table 2, Table 3, Table 4, and Table 5. The authors might report the fitness evaluation of the model.

13. Page 20, Line 281-282. The post PCV13 carriage survey showed increased pneumococcal density. Except the reference 16 (from the same study cohort as the present study), are there supportive references? or are there possible explanations? The authors might discuss the finding in detail.

Reviewer #3: This study examined the prevalence and density of nasopharyngeal pneumococcal carriage in children of the Lao People's Democratic Republic before and after the introduction of the 13-valent pneumococcal conjugate vaccine (PCV13).

Nasopharyngeal swabs were collected infants from aged 5 - 8 weeks old (n = 999) and children aged 12 - 23 months (n = 1,010), pneumococci detected by quantitative PCR, and a risk factor questionnaire completed. Logistic and linear regression models were used to evaluate associations between participant characteristics and pneumococcal carriage and density.

In infants aged 5 - 8 weeks, living in a household with two or more children under the age of five years (aOR 1.97; 95% CI 1.39 - 2.79) and low family income (aOR 1.64; 95% CI 0.99 - 2.72) were positively associated with pneumococcal carriage. For children aged 12 - 23 months, upper respiratory tract infection (URTI) symptoms, two or more children under five in the household, and rural residence were positively associated with pneumococcal carriage. PCV13 vaccination was negatively associated with carriage of PCV13 serotypes. URTI symptoms, current breastfeeding, rural residence, and delivery by Caesarean section were associated with higher mean pneumococcal density in pneumococcal carriers.

Overall, this study provides new data on nasopharyngeal carriage in a developing country in South East Asia. Although the methodology is not novel, this study provides new data on pneumococcal burden setting in southeast Asia.

One point needs more elaboration. In this study, current breastfeeding was associated with higher mean pneumococcal density in pneumococcal carriers. This is in contrast to most previous examples about the effect of breast feeding on microbial infection of young children. Although some explanations were raised in the Discussion, more in-depth discussion on the biological basis should be considered.

6. PLOS authors have the option to publish the peer review history of their article (what does this mean?). If published, this will include your full peer review and any attached files.

Reviewer #1: No

Reviewer #2: No

Reviewer #3: No

---

## [Author Response · Author response to Decision Letter 0]

1 Sep 2019

The authors thank the reviewers for the time that they put into reviewing our manuscript and for their thoughtful comments and feedback. Line numbers included in our responses refer to the revised version of the manuscript.

Review Comments to the Author

Reviewer #1: In this manuscript, the authors analyzed the relationship between host factors and pneumococcal carriage density in the young children in the Lao People's Democratic Republic. They provide evidence that some host factors, including living environments, areas, socio-economic factors and URTI infections, would influence the pneumococcal carriage and density in young children. These data provide some information on pneumococcal carriage and density in a high pneumococcal disease burden in Lao, but some important information should be provided more clearly. The details of my concerns are listed as follows.

1. To quantify the bacterial density, the authors use the unit of “genome equivalents per ml (GE/ml)” to represent the bacterial density in this study, but little information about the unit of GE/ml was obtained from this manuscript. The authors should describe how to detect the “genome equivalents per ml” in more detail and tell us what the volume unit represent. They should also provide evidence or reference that the data of “genome equivalents per ml” can represent the absolute number of pneumococci isolated from the nasopharyngeal swab samples. The distribution of bacteria number isolated form the clinical samples are always wide. The result of the pneumococcal density showed in this study is only arranged from 5.49 to 6.00 (log10 GE/ml), which implied the limitation of the quantification assay. Therefore, more detailed description and evidence should be provided to support the presenting data.

Authors’ response: 

We have provided added information on the definition of GE/ml and how it is calculated, as well as more details on the lytA qPCR assay as suggested by Reviewer #2 (comment 5) to the methods section in lines 112 - 119. Consistent with other studies, we also observed a wide distribution of pneumococcal density, ranging from 2.9 to 8.2 log10 GE/ml. We have added these data to the results section in lines 179 - 181, and noted the wide distribution in the discussion (lines 301 - 302). Note that only mean densities are presented in Table 5 in the interest of space and do not represent the full range of densities observed. 

2. PVC vaccination will significantly reduce the carriage of PCV13 serotypes (Table 4). Therefore, the carriage of pneumococci or PCV13 serotypes in the group of pre-vaccination should be specifically analyzed.

Authors’ response: 

We conducted an additional analysis as suggested to examine carriage of PCV13 serotypes in 12-23 month old children specifically in the pre-PCV survey. Results were consistent with the original results (from both surveys combined) shown in Table 4, with rural residence, upper respiratory tract symptoms, and living in a household with two or more children under the age of five years found to be associated with PCV13 serotype carriage. The results of this additional analysis have been included in the results section in lines 219 - 226. 

3. Some results described in the abstract, in the text and showed in the able are not consistent, such as the association of living in a household with two or more children under the age of five years (infants aged 5 - 8 weeks) and pneumococcal carriage. Please check that carefully.

Author’s response: 

Thank you for that observation. We have checked all of the data and fixed the error identified in the results text as well as a couple of other typos identified. 

Reviewer #2: The authors identified risk factors for pneumococcal carriage and density using data from cross-sectional pneumococcal carriage surveys conducted in the Lao People's Democratic Republic before and after the introduction of the 13-valent pneumococcal conjugate vaccine (PCV13). The authors found that in the age group of 5-8 weeks, having and two or more children under five in the 233 household and low family income were risk factors for pneumococcal carriage. Children aged 12 - 23 months, URTI symptoms, two or more children under five years old in the household, and rural residence were associated with pneumococcal carriage. URTI symptoms, current breastfeeding, rural residence, and delivery by Caesarean section were associated with higher mean pneumococcal density in pneumococcal carriers.

I have some major concerns.

1. The authors have published the other article using the same cohort. (Vaccine. 404 2019;37(2):296-305.) The author should clearly state the difference between the two studies.

Authors’ response: 

The objective of the original study published in Vaccine was to evaluate the impact of PCV13 introduction on pneumococcal carriage and circulating serotypes. Specifically, we compared carriage rates pre- and 2 years post PCV13 introduction. The objective of this current manuscript was to identify demographic and household risk factors for pneumococcal carriage and pneumococcal carriage density. We have added the following information to the introduction (lines 75 - 77) to clearly highlight the differences between these two studies:

“Here, we describe a secondary analysis conducted using data from these cross-sectional surveys to identify demographic and household factors associated with carriage of pneumococci overall, PCV13 serotype carriage, and pneumococcal carriage density.”

2. The pneumococcal density was measured using real-time PCR according to the reference (J Clin Microbiol. 2007;45(8):2460-6.). However, in the initial study by Carvalho, the authors used the real-time PCR to detect rather than quantify the pneumococcus. Several studies are using the same method to quantify the pneumococcus. However, there are augment that measuring the pneumococcus DNA alone is likely to be confounded by variable production and sampling of nasopharyngeal mucous. (Sci Rep. 2018 Jul 23;8(1):11030.). I am wondering in the present study whether the pneumococcal density adjusted for the concentration of human DNA present? If "no", the authors might consider to debate why the adjustment is not necessary.

Authors’ response: 

We did not adjust for human DNA concentration in this study. Our laboratory has considered whether this approach would be useful for pneumococcal carriage studies, and upon careful consideration we determined that it would not be of sufficient benefit to implement. As human DNA greatly exceeds pneumococcal DNA in a swab sample, the alu qPCR described in the 2018 Sci Reports paper requires a 1:1000 dilution of template DNA. This necessitates additional manual handling of specimens, which is a risk for contamination, particularly cross-contamination between samples. Running a second qPCR assay would also incur additional labor and consumables costs. Notably, the authors of this paper report that “the adjustment made little difference to the results, suggesting that the nasopharyngeal flocked swab is efficient at collecting a standard volume of secretion from the posterior nasopharyngeal mucosa.” As the adjustment made little difference to the results, while increasing the laboratory workload and risk of contamination, we chose not to implement this procedure. To help ensure consistent sampling of the nasopharynx, swab collection was conducted by trained study personnel, performed using flocked swabs, and swab collection, transport, and storage were conducted in accordance with WHO recommendations (Vaccine. 2013;32(1):165-79).

There also some additional comments.

1. If the data can be delinked, what is the ethical concerns that the data can't be publicly available?

Authors’ response:

The ethical concerns preventing making data publicly available are not due to personally identifying information, which as noted, can be delinked. Rather, the issue is that the protocol that was approved by ethics committees in Lao PDR and Australia specified the purpose of the study and what data would be used for, and this information was also conveyed in the informed consent process. Therefore, reasonable requests for data would need to include details on how the data would be used and will be subject to approval by the Lao PDR Ministry of Health National Ethics Committee for Health Research, which is the overseeing ethics committee for this study. Additionally, this process is mindful of potential sensitivities regarding data from ethnic minorities.

2. Page 2 Line 30. The variable low family income seems statistically not significant.

Authors’ response:

In keeping with the recent movement away from using p < 0.05 as a strict cutoff for statistical significance (see Wasserstein and Lazar. The ASA Statement on p-Values: Context, Process, and Purpose. The American Statistician 2016 70(2): 129-133; Harrington et al. New Guidelines for Statistical Reporting in the Journal. N Engl J Med. 2019 Jul 18;381(3):285-286.) throughout the manuscript we have presented p values and 95% confidence intervals (CI) and taken these measures as well as the odds ratio into account when interpreting results, and avoided describing our results as ‘statistically significant’. Our interpretation is that the adjusted odds ratio of 1.64 (95% CI 1.39 – 2.79) provides some evidence of a positive association, as described in the results (lines 185 - 186). In the abstract this description has been shortened however the reader can interpret the data using the 95% CI.

3. Page 4 Line 73-75. The statement is statistically not significant.

Authors’ response:

These results are described as showing “some evidence of indirect effects” and not statistically significant in the text, with an explanation provided above.

4. Study design and participant section. In what healthcare setting were those participant enrolled? Were the participant of both age group enrolled in the same healthcare setting?

Authors’ response:

 Participants of both age groups were primarily enrolled during routine clinic visits, however some rural participants were enrolled via household visits. We have added this information to the relevant section of the methods (lines 84 - 85).

5. page 6, Line 107-110. The author might describe more detail about the method of the realtime PCR. What is used as a standard curve? How the density calculate? What is the detection limits? Did the experiment include positive control or negative control of the swab? Or did the Real-time PCR included positive or negative control?

Authors’ response: In response to these questions and those of Reviewer #1 (comment 2), we have added more information about the qPCR assay, standard curve, controls, and how density is calculated to the methods section (lines 112 – 119).

6. Page 7, Line 132. Is there a preset cutoff for the statistical significance?

Authors’ response: As described in the response to comment 2, we did not use a preset cutoff for statistical significance. 

7. Page 7, Line 144. According to table 1, there are 507 participants of the age group 12-23 months in the two years post-PCV 13 . Why the denominator is 497 in Line 144 ?

Authors’ response: 

Information on vaccine history was unable to be verified from written records for 10 participants. We have added this information to the results in lines 156 – 157.

 Table 1. why the denominator varies in different rows?

Authors’ response: 

Some data were missing for particular variables, therefore these participants were excluded when percentages were calculated for that variable. We have edited the footnote to explain this. 

9. if the participant of both age group enrolled in the same healthcare setting, it is weird that there are large differences of the distribution of the rural residence and the URTI symptoms.

Authors’ response: 

We agree that this is a noticeable difference. As for the low proportion of infants aged 5 – 8 weeks enrolled in rural areas, we recruited in one province over a relatively short time frame, and due to a lower populations size and number of births, the number of potentially eligible participants in this age group was smaller than those residing in urban areas. We have mentioned the low proportion of 5 – 8 week olds from rural areas in the discussion (lines 275 - 277).

In terms of URTI symptoms, results are consistent with studies conducted by us and others showing that URTI symptoms are more common in older infants and toddlers than very young infants (Kusel et al. Pediatr Infect Dis J. 2006 Aug;25(8):680-6; Dunne et al. Lancet Glob Health. 2018 Dec;6(12):e1375-e1385.) The most likely explanation for this is the lower exposure of very young infants to respiratory viruses, as they are more likely to be kept indoors and typically do not attend school or day care. Traditionally in Lao PDR, for the first month of an infant’s life, both mother and baby stay within the home, with limited visitors. These practices would reduce exposure to URTIs in the young infant age group.

10. Page 8, Line 155-158. Why are the denominators different? 999 or 995, and 1009 or 1005?

Authors’ response:

The denominators are different as specimens that were not available for serotyping were excluded, therefore the denominators differed by group depending on the number of available samples. We have added a sentence in the methods noting that when serotyping results were not available due to technical issues, that sample was excluded from serotype-specific analyses (lines 142 - 143).

11. Page 9, Line 164-165. The statement is statistically not significant.

Authors’ response: 

The adjusted odds ratio of 1.64 (95% CI 1.39 – 2.79) is described as ‘some evidence for a positive association’ in line with our interpretation of these data, as explained in response to comment 2.

As noted in 

12. Table 2, Table 3, Table 4, and Table 5. The authors might report the fitness evaluation of the model.

Authors’ response:

We have added the pseudo R2 or adjusted R2 values as appropriate to table footnotes. 

13. Page 20, Line 281-282. The post PCV13 carriage survey showed increased pneumococcal density. Except the reference 16 (from the same study cohort as the present study), are there supportive references? or are there possible explanations? The authors might discuss the finding in detail.

Authors’ response: 

Published data on the effects of PCV and pneumococcal density have shown varying results. As suggested, we have expanded the discussion and cited additional references (lines 309 - 318).

Reviewer #3: This study examined the prevalence and density of nasopharyngeal pneumococcal carriage in children of the Lao People's Democratic Republic before and after the introduction of the 13-valent pneumococcal conjugate vaccine (PCV13).

Nasopharyngeal swabs were collected infants from aged 5 - 8 weeks old (n = 999) and children aged 12 - 23 months (n = 1,010), pneumococci detected by quantitative PCR, and a risk factor questionnaire completed. Logistic and linear regression models were used to evaluate associations between participant characteristics and pneumococcal carriage and density.

In infants aged 5 - 8 weeks, living in a household with two or more children under the age of five years (aOR 1.97; 95% CI 1.39 - 2.79) and low family income (aOR 1.64; 95% CI 0.99 - 2.72) were positively associated with pneumococcal carriage. For children aged 12 - 23 months, upper respiratory tract infection (URTI) symptoms, two or more children under five in the household, and rural residence were positively associated with pneumococcal carriage. PCV13 vaccination was negatively associated with carriage of PCV13 serotypes. URTI symptoms, current breastfeeding, rural residence, and delivery by Caesarean section were associated with higher mean pneumococcal density in pneumococcal carriers.

Overall, this study provides new data on nasopharyngeal carriage in a developing country in South East Asia. Although the methodology is not novel, this study provides new data on pneumococcal burden setting in southeast Asia.

One point needs more elaboration. In this study, current breastfeeding was associated with higher mean pneumococcal density in pneumococcal carriers. This is in contrast to most previous examples about the effect of breast feeding on microbial infection of young children. Although some explanations were raised in the Discussion, more in-depth discussion on the biological basis should be considered.

 Authors’ response: 

We have added some additional possible explanations of this result to the discussion as suggested (lines 335 - 341). These include possible confounding- there may something different about infants breast fed for longer that is also associated with higher pneumococcal density. One plausible biological explanation could due to indirect effects- breast feeding has an inhibitory effect on a bacterium such as S. aureus which is a competitor of pneumococcus.

---

## [Decision Letter · Decision Letter 1]

17 Sep 2019

PONE-D-19-19269R1

Factors associated with pneumococcal carriage and density in infants and young children in Laos PDR

PLOS ONE

Dear Dr Dunne,

Thank you for submitting your manuscript to PLOS ONE. After careful consideration, we feel that it has merit but does not fully meet PLOS ONE’s publication criteria as it currently stands. Therefore, we invite you to submit a revised version of the manuscript that addresses the points raised during the review process.

We would appreciate receiving your revised manuscript by Nov 01 2019 11:59PM. To enhance the reproducibility of your results, we recommend that if applicable you deposit your laboratory protocols in protocols.io, where a protocol can be assigned its own identifier (DOI) such that it can be cited independently in the future. For instructions see: http://journals.plos.org/plosone/s/submission-guidelines#loc-laboratory-protocols

We look forward to receiving your revised manuscript.

Kind regards,

Jean-San Chia

Academic Editor

PLOS ONE

Additional Editor Comments (if provided):

Please address the concerns of reviewer one. [The authors have addressed all my concerns, but I still have a minor suggestion. The range of each data of bacterial density can be provided to make the presenting data more clear.]

Reviewers' comments:

Reviewer's Responses to Questions

**Comments to the Author**

1. If the authors have adequately addressed your comments raised in a previous round of review and you feel that this manuscript is now acceptable for publication, you may indicate that here to bypass the “Comments to the Author” section, enter your conflict of interest statement in the “Confidential to Editor” section, and submit your "Accept" recommendation.

Reviewer #1: All comments have been addressed

Reviewer #2: All comments have been addressed

2. Is the manuscript technically sound, and do the data support the conclusions?

Reviewer #1: Yes

Reviewer #2: Yes

3. Has the statistical analysis been performed appropriately and rigorously? 

Reviewer #1: Yes

Reviewer #2: Yes

4. Have the authors made all data underlying the findings in their manuscript fully available?

Reviewer #1: Yes

Reviewer #2: No

5. Is the manuscript presented in an intelligible fashion and written in standard English?

Reviewer #1: Yes

Reviewer #2: Yes

6. Review Comments to the Author

Reviewer #1: The authors have addressed all my concerns, but I still have a minor suggestion. The range of each data of bacterial density can be provided to make the presenting data more clear.

Reviewer #2: (No Response)

7. PLOS authors have the option to publish the peer review history of their article (what does this mean?). If published, this will include your full peer review and any attached files.

Reviewer #1: No

Reviewer #2: No

---

## [Author Response · Author response to Decision Letter 1]

3 Oct 2019

The authors thank the editor and reviewers for their consideration of our revised manuscript and the feedback provided. Please find a response to reviewer comments below. Line numbers refer to the clean, final version of the manuscript.

Review Comments to the Author

Reviewer #1: The authors have addressed all my concerns, but I still have a minor suggestion. The range of each data of bacterial density can be provided to make the presenting data more clear.

Authors’ response:

As suggested, we have added ranges of bacterial density data in the results text (lines 180 – 181) and to Table 5.

---

## [Decision Letter · Decision Letter 2]

14 Oct 2019

Factors associated with pneumococcal carriage and density in infants and young children in Laos

PONE-D-19-19269R2

Dear Dr. Dunne,

We are pleased to inform you that your manuscript has been judged scientifically suitable for publication and will be formally accepted for publication once it complies with all outstanding technical requirements.

With kind regards,

Jean-San Chia

Academic Editor

PLOS ONE

Additional Editor Comments (optional):

Reviewers' comments:

Reviewer's Responses to Questions

**Comments to the Author**

1. If the authors have adequately addressed your comments raised in a previous round of review and you feel that this manuscript is now acceptable for publication, you may indicate that here to bypass the “Comments to the Author” section, enter your conflict of interest statement in the “Confidential to Editor” section, and submit your "Accept" recommendation.

Reviewer #1: All comments have been addressed

2. Is the manuscript technically sound, and do the data support the conclusions?

Reviewer #1: Yes

3. Has the statistical analysis been performed appropriately and rigorously? 

Reviewer #1: Yes

4. Have the authors made all data underlying the findings in their manuscript fully available?

Reviewer #1: Yes

5. Is the manuscript presented in an intelligible fashion and written in standard English?

Reviewer #1: Yes

6. Review Comments to the Author

Reviewer #1: The authors have addressed all my concerns. It has been a well-written manuscript. I have no other comments.

7. PLOS authors have the option to publish the peer review history of their article (what does this mean?). If published, this will include your full peer review and any attached files.

Reviewer #1: No

---

## [Editor Report · Acceptance letter]

21 Oct 2019

PONE-D-19-19269R2 

Factors associated with pneumococcal carriage and density in infants and young children in Laos PDR 

Dear Dr. Dunne:

I am pleased to inform you that your manuscript has been deemed suitable for publication in PLOS ONE. Congratulations! Your manuscript is now with our production department. 

With kind regards,

on behalf of

Dr. Jean-San Chia 

Academic Editor

PLOS ONE